# The Role of *Epinephelus coioides* DUSP5 in Regulating Singapore Grouper Iridovirus Infection

**DOI:** 10.3390/v15091807

**Published:** 2023-08-25

**Authors:** Jiayang He, Yijie Cai, Wei Huang, Yunxiang Lin, Yurong Lei, Cuifen Huang, Zongbin Cui, Qiwei Qin, Hongyan Sun

**Affiliations:** 1State Key Laboratory of Applied Microbiology Southern China, Institute of Microbiology, Guangdong Academy of Sciences, Guangzhou 510070, China; hejiayang1997@sina.com; 2Guangdong Laboratory for Lingnan Modern Agriculture, College of Marine Sciences, South China Agricultural University, Guangzhou 510642, China; caiyijie19980802@126.com (Y.C.); 2421311735hw@gmail.com (W.H.); 201929210311@stu.scau.edu.cn (Y.L.); 13750339910@163.com (Y.L.); 13432285679@163.com (C.H.); 3Southern Marine Science and Engineering Guangdong Laboratory, Zhuhai 519000, China; 4Laboratory for Marine Biology and Biotechnology, Qingdao National Laboratory for Marine Science and Technology, Qingdao 266000, China

**Keywords:** *Epinephelus coioides*, DUSP5, SGIV, immune response, apoptosis

## Abstract

The dual-specificity phosphatase (DUSP) family plays an important role in response to adverse external factors. In this study, the DUSP5 from *Epinephelus coioides*, an important marine fish in Southeast Asia and China, was isolated and characterized. As expected, *E. coioides* DUSP5 contained four conserved domains: a rhodanese homology domain (RHOD); a dual-specificity phosphatase catalytic domain (DSPc); and two regions of low compositional complexity, indicating that *E. coioides* DUSP5 belongs to the DUSP family. *E. coioides* DUSP5 mRNA could be detected in all of the examined tissues, and was mainly distributed in the nucleus. Infection with Singapore grouper iridovirus (SGIV), one of the most important pathogens of marine fish, could inhibit the expression of *E. coioides* DUSP5. The overexpression of DUSP5 could significantly downregulate the expression of the key SGIV genes (MCP, ICP18, VP19, and LITAF), viral titers, the activity of NF-κB and AP-I, and the expression of pro-inflammatory factors (IL-6, IL-8, and TNF-α) of *E. coioides*, but could upregulate the expressions of caspase3 and p53, as well as SGIV-induced apoptosis. The results demonstrate that *E. coioides* DUSP5 could inhibit SGIV infection by regulating *E*. *coioides* immune-related factors, indicating that DUSP5 might be involved in viral infection.

## 1. Introduction

Dual-specificity phosphatase (DUSP) has the ability to dephosphorylate both serine/threonine and tyrosine residues on their substrate protein. It also plays important roles in cell proliferation, cell survival, apoptosis, and the body’s resistance to adverse external factors [1,2]. All DUSPs possess the conserved domains of Asp and Cys, as well as the catalytic sites associated with arginine residue formation. DUSPs containing a KIM structural domain are usually classified as typical DUSPs, whereas DUSPs without a KIM structural domain are usually classified as atypical DUSPs [3,4].

DUSP5, a member of DUSPs, is a cellular phosphatase, and can regulate MAPK family via dephosphorylating both the phosphoserine/threonine and phosphor-tyrosine residues [5]. In mammals, DUSP5 can usually be ubiquitinated or degraded by the proteasome after its action [5]. DUSP5 is also an important influential factor of the MAPK signaling pathway, and can regulate a variety of intracellular responses, such as inflammation [6,7,8,9]. DUSP5 usually exhibits phosphatase activity against ERK of the MAPK superfamily, contributing to ERK inactivation and nuclear translocation [10]. Overexpressed DUSP5 could inhibit the formation of autophagic vesicles by inhibiting the phosphorylation of signaling molecules in the signaling pathway of ERK1/2, and the knockdown of DUSP5 could increase the expression of autophagy-associated proteins (Atgs) [11,12]. DUSP5 can negatively regulate the signaling pathways of ERK and NF-κB in order to affect the bacterial infection-induced inflammatory responses [13]. The knockdown of DUSP5 could promote MYXV viral replication, which is related to the similarity of DUSP5 to MYXV 069L [14]. The E7 protein of human papillomavirus type 16 (HPV16) could inhibit the expression of DUSP5, which causes the activation of MAPK/ERK signaling and induction of cell autophagy by mTOR and MAPK [15]. In *Paralichthys olivaceus*, DUSP5 led to a differential expression response to LPS and polyI: C stimulation [3]. However, the role of DUSP5 in viral infection of teleosts remains need further study.

The grouper *Epinephelus coioides* belongs to *Perciformes*, *Serranidae*, and is an important marine fish with great economic value [16]. The relatively high farming density has also caused outbreaks of fish diseases, and the increasingly frequent occurrence of fish diseases has seriously affected the sustainable development of the grouper farming industry. Among them, Singapore grouper iridovirus (SGIV) is one of the important pathogens causing grouper disease, with a mortality rate of over 90%, and leads to significant economic losses in the aquaculture industry [17,18]. Based on this information, the aim of this study was to characterize *E. coioides* DUSP5, study its expression pattern and cell distribution, and elucidate the role of *E. coioides* DUSP5 during SGIV infection.

## 2. Material and Methods

### 2.1. Experimental Fish, Cells, and Virus

A total of 200 healthy *E. coioides* (64.8 g ± 4.02 g) from a local fish farm in Huizhou, Guangdong Province, People’s Republic of China, were cultured using a circulating water system and 30‰ salinity at 27 °C for two weeks before experimentation [19]. Twenty fish were randomly selected to determine the dose of infection for preliminary experiments. The remaining 180 fish were randomly grouped in triplicate with 30 fish per tank for SGIV infection, and SGIV used in this study was maintained at −80 °C, as described previously [19]. The untreated fish were used as the control group. *E. coioides* were injected intraperitoneally with 100 μL SGIV (6.23 × 10^6^ TCID_50_/_mL_) [19]. Three fish per tank were randomly sampled at 0, 3, 6, 12, 24, 48 and 72 h after injection. Subsequently, 12 tissues, including the liver, spleen, intestine, stomach, trunk kidney, blood, heart, gill, muscle, skin, head kidney, and brain were sampled at −80 °C for further analysis. 

The grouper spleen (GS) cells and fathead minnow (FHM) cells used here were cultured at 28 °C in Leibovitz’s L15 medium containing 10% fetal bovine serum (Gibco, Rockville, MD, USA) [19]. GS cells in 24-well plates were infected with SGIV and collected randomly for three wells at 0 h, 2 h, 4 h, 6 h, 10 h, 12 h, 24 h, 36 h, and 48 h [19]. FHM cells were used for SGIV-induced apoptosis.

### 2.2. Total RNA Extraction and cDNA Synthesis

Total RNA was isolated from the tissues/cells using TRIzol reagent (Invitrogen, Burlington, ON, Canada), and the first-strand cDNA was synthesized using Rever Tra Ace-a reverse transcriptase (Toyobo, Japan). The cDNA was used as a template for the following polymerase chain reaction (PCR) (TaKaRa) and quantitative real-time PCR (qPCR) test (Thermo Scientific, Waltham, MA, USA). 

### 2.3. Identification of E. coioides DUSP5

The expressed sequence tag (EST) sequence of *E. coioides* DUSP5 was identified from the transcriptome database that was obtained by our lab. The primers, i.e., DUSP5-ORF-F/R (Table 1), were designed to amplify DUSP5 cDNA. PCR was performed in a final volume of 100 μL, including 4 μL template DNA, 4 μL of each primer, 50 μL PrimeSTAR HS (Premix) (TaKaRa, Japan), and 38 μL H_2_O. The conditions were as follows: 34 cycles of 98 °C, 10 s; 98 °C, 10 s; 55 °C, 5 s, and 72 °C, 75 s, followed by 72 °C for 5 min. The PCR products were sequenced by Sangon Biotech (Shanghai, China). The PCR products were purified and cloned into pEGFP-C1/pcDNA3.1-3HA using ClonExpress Ulira One Step Cloning Kit (Vazyme, Nangjing, China), and the recombinant plasmid (pEGFP-DUSP5 and pcDNA3.1-DUSP5) was confirmed by DNA sequencing. 

### 2.4. Biological Information

The ORF of *E. coioides* DUSP5 was found by NCBI ORF finder tool (https://www.ncbi.nlm.nih.gov/orffinder/, accessed on 1 May 2023). The conserved domain was searched in (http://www.ncbi.nlm.nih.gov/Structure/cdd/wrpsb.cgi, accessed on 1 May 2023). The isoelectric point (pI) and molecular weight (Mw) of *E. coioides* DUSP5 were calculated with pI/Mw tool (http://web.expasy.org/compute_pi/, accessed on 1 May 2023). The amino acid sequence of *E. coioides* DUSP5 was compared with other species using the CLUSTALW server (http://www.ebi.ac.uk/clustalw, accessed on 1 May 2023). The domain and location were predicted by means of the SMART tool (http://smart.embl-heidelberg.de/, accessed on 1 May 2023). A neighbor-joining phylogenic tree based on the sequences of the amino acids was created using MEGA 5.04, and the bootstraps were set as 1000.

### 2.5. The Expression Patterns of E. coioides DUSP5

The expression patterns of *E. coioides* DUSP5 were assessed using quantitative real-time PCR amplification (qPCR), and the cDNA was used as a template. The primers are listed in Table 1. The primers DUSP5- RT-F and DUSP5-RT-R were designed for qPCR, and β-actin was used as the reference gene. qPCR was performed with SYBR Green Real-Time PCR Master Mix (Toyobo) using an Applied Biosystems QuantStudio 5 Real Time Detection System (Thermo Fisher, Waltham, MA, USA). The conditions were 1 cycle of 95 °C for 1 min followed by 40 cycles of 95 °C, 15 s; 60 °C, 15 s; 72 °C, 45 s. The expression of the target gene was normalized to the reference gene and calculated with the 2^−ΔΔCt^ method. 

### 2.6. Subcellular Localization

To elucidate the subcellular localization of *E. coioides* DUSP5, GS cells were cultured in a 33 mm dish for 18 h before transfection. The plasmids pEGFP-C1 (800 ng, control) and pEGFP-C1-*E. coioides* DUSP5 (800 ng) were transfected into GS cells for 24 h using the lipofectamine™ 2000 reagent (Invitrogen, Waltham, MA, USA), respectively. The cells were fixed with 4% paraformaldehyde for 15 min, washed with 1 × PBS three times, and stained with 4, 6-diamidino-2-phenylindole (DAPI) (Bioforxx, Einhausen, Germany) for 10 min. The position of the protein labeled by green fluorescent was detected using the fluorescence microscope.

### 2.7. Western Blotting

GS cells transfected with 800 ng pcDNA-3.1-3HA/pcDNA-3.1-DUSP5 for 24 h were infected with SGIV for 24 h. The cells were collected and lysed by pierce IP lysis buffer (Thermo Fisher), then separated by 10% sodium dodecyl sulfate–polyacrylamide gel electrophoresis (SDS-PAGE). Subsequently, the proteins were transferred onto 0.2 μm PVDF membranes (Millipore, Billerica, MA, USA) and blocked using 5% skim milk for 2 h. The membranes were incubated with rabbit anti-MCP antibody (1:1000 dilution), which was prepared in our lab with the patent number CN111363758A [19]. Rabbit anti-β-tubulin antibody (1:2000 dilution), anti-3HA antibody (1:1000 dilution), anti-caspase3 (1:1000 dilution), and anti-cleaved caspase3 (1:1000 dilution) were purchased from Abcam. The horseradish peroxidase (HRP)-conjugated goat anti-rabbit (1:5000 dilution) was purchased from Abcam as well. Immuno-reactive proteins were visualized by an enhanced HRP-DAB Chromogenic Substrate Kit (Tiangen, Beijing, China). 

### 2.8. Virus Titer

To determine the role of DUSP5 during SGIV infection, TCID_50_ was used to evaluate the viral titer. GS cells transfected with the plasmids pcDNA-3.1-3HA/pcDNA-3.1-DUSP5 for 24 h were infected with SGIV for 24 h, collected, and freeze-thawed three times at −80 °C. The cell lysate was diluted tenfold and used to infect GS cells in 96-well plates. About 5 days after infection, the viral titer was calculated using TCID_50_.

### 2.9. Dual-Luciferase Reporter

To detect the effect of DUSP5 on the activity of the NF-κB/AP-I promoter after SGIV infection, luciferase reports containing NF-κB/AP-I were used for co-transfection. In detail, GS cells in 24-well plates were transfected with the luciferase plasmids (NF-κB-Luc/AP-1-Luc, 150 ng/well), internal control pRL-SV40 Renilla luciferase vector (40 ng/well), and pcDNA3.1-3HA/pcDNA3.1-3HA-DUSP5 (800 ng/well) using Lipofectamine 2000, respectively. After 24 h, the cells were infected with SGIV for 48 h, then harvested. The cell lysates were collected using passive lysis buffer, and the luciferase activities were measured using the Dual-Luciferase^®^ Reporter Assay System (Promega, Madison, WI, USA).

### 2.10. Cell Apoptosis

To explore the function of DUSP5 in cell apoptosis during SGIV infection, FHM cells in 24-well plates were transfected with 800 ng pcDNA-3.1-3HA/pcDNA-3.1-DUSP5 for 24 h using Lipofectamine 2000, then infected by SGIV for 24 h. The cells were harvested, and apoptosis was detected by flow cytometry using the annexin VFITC apoptosis detection kit (Beyotime, Chengdu, China). Each experiment was conducted in triplicate. Data acquisition and analysis were performed using a flow cytometry system (Beckman Coulter, Indianapolis, IN, USA) and FlowJo VX software v10.9.

### 2.11. Statistical Analysis 

All of the data were expressed as means ± standard error of the mean (SD) from the separate experiments and analyzed with GraphPad Prism 7.0 using one-way ANOVA, followed by Duncan’ s test. Significance was set at *p* < 0.05.

## 3. Result

### 3.1. Molecular Characterization of E. coioides DUSP5

The full-length cDNA sequence of *E. coioides* DUSP5 was obtained, and we submitted it to GenBank under accession no. OQ054325. 

*E. coioides* DUSP5 is 1638 bp in length, including a 126 bp 5’ UTR; a 381 bp 3’ UTR; and a 1131 bp open reading frame (ORF) encoding 376 amino acids, with a molecular weight of 42.11 kDa. The instability index (II) is 61.33, which classifies the protein as unstable, and the grand average of hydropathicity (GRAVY) is −0.257, suggesting it is a hydrophilic protein (Figure 1A). *E. coioides* DUSP5 has 43 phosphorylation sites: 32 serine sites, 8 threonine sites, and 3 tyrosine sites, which include 5 MAPK phosphorylation sites located in Ser293, Thr319, Thr324, Ser361, and Ser370. In addition, *E. coioides* DUSP5 has two N glycosylation sites in Asn35 and Asn202 (Figure 1A). In terms of structural domains, DUSP5 contains a rhodanese homology domain (RHOD) (aa8-133), a dual-specificity phosphatase, a catalytic domain (DSPc) (aa170-308), and two regions of low compositional complexity (aa312-325, aa333-347). Tertiary structure predictions indicate a spatial structure dominated by spirals and irregular spins. The five predicted p38 MAPK phosphorylation sites were located at the Ser293, thr319, Thr324, ser361, and Ser370 sites, with Ser293 located in the conserved DSPc structural domain. As shown in Figure 1B, *E. coioides*, *E. lanceolatus*, *Miichthys miiuy*, *Mastacembelus armatus,* and *Danio rerio* DUSP5 all had five p38 MAPK phosphorylation sites, and they each had an identical serine site located in the conserved DSPc structural domain. 

Phylogenetic analysis showed that DUSP5 from different species clustered into two branches: teleosts clustered into one group and mammals, birds, and reptiles clustered into another branch. *E. coioides* DUSP5 was divided into the teleost, and it had the highest amino acid identity with *E. lanceolatus* and the lowest amino acid identity with *Mus caroli* (Figure 1C).

### 3.2. Expression and Localization of E. coioides DUSP5

To examine the tissue distribution of *E. coioides* DUSP5, the total RNA of 12 tissues from healthy *E. coioides* was extracted, and qPCR was used to quantify the expression. As shown in Figure 2A, *E. coioides* DUSP5 mRNA was detected in all of the examined tissues, but the expression levels were different: the expression level was higher in the skin, followed by the brain, gill, muscle, heart, liver, trunk kidney, spleen, head kidney, stomach, intestine, and blood. 

The subcellular localization of genes has great significance for studying their functions [20]. In this study, an *E. coioides* DUSP5 ORF sequence was constructed into a pEGFP-C1 plasmid with green fluorescent protein and transfected into GS cells. As shown in Figure 2B, *E. coioides* DUSP5 was distributed in the nucleus of GS cells, compared with the control group.

### 3.3. Differential Expression of E. coioides DUSP5 during SGIV Infection

The expression levels of DUSP5 mRNA were obtained during SGIV infection by qPCR. After SGIV infection, the expression of DUSP5 was downregulated at 3 h in both the liver and spleen, but upregulated at 6 h, at which time a peak occurred in both the liver (5.43-fold, *p* < 0.05) and spleen (1.32-fold, *p* < 0.05), and was then downregulated again (Figure 3).

### 3.4. Overexpression Efficiency of E. coioides DUSP5

DUSP5 ORF was constructed into the pcDNA-3.1-3HA vector, and then transfected into GS cells using Lipofectamine 2000. At 12 h, 24 h, and 48 h, the cells were collected and RNA was extracted by TRIzol reagent. As shown in Figure 4A, overexpression of pcDNA-3.1-3HA-DUSP5 in GS cells reached a peak (3182.4-fold) at 24 h after transfection compared to the control group, according to qPCR. By Western blotting, HA protein could be detected at about 45 kDa after transfection with pcDNA-3.1-3HA-DUSP5 for 24 h (Figure 4B). The results indicate that, after transfection with pcDNA-3.1-3HA-DUSP5, both the mRNA and protein of *E. coioides* DUSP5 could be transcribed and translated normally in GS cells.

### 3.5. E. coioides DUSP5 Inhibits SGIV Replication

To explore the function of DUSP5 during SGIV infection, the GS cells transfected using pcDNA-3.1-3HA-DUSP5 for 24 h were infected with SGIV for 24 h. The CPE formed by SGIV infection, the expression of key SGIV genes, and the protein level of SGIV-MCP were studied. As shown in Figure 5, overexpressing DUSP5 can reduce SGIV-induced CPE compared to the control group (Figure 5A). DUSP5 was able to significantly inhibit the expressions of key SGIV genes MCP, ICP18, VP19, and LITAF, according to qPCR (*p* < 0.05) (Figure 5B). By Western blotting, DUSP5 was able to reduce the expression of SGIV-MCP at the protein level (Figure 5C). The viral titer of DUSP5-overexpressing cells was significantly lower than that of the control cells (Figure 5D). These results indicate that the overexpression of DUSP5 could inhibit SGIV replication.

### 3.6. E. coioides DUSP5 Negatively Regulates Innate Immunity

In order to further explore the mechanism of DUSP5 inhibiting SGIV infection, the effect of DUSP5 on host cell inflammation was evaluated by qPCR or luciferase reporter gene assay. The expression levels of inflammatory factors (IL-6, Il-8, and TNF-α) in DUSP5-overexpression GS cells were significantly downregulated during SGIV infection by qPCR (Figure 6A). Due to the luciferase reporter gene activity experiment, the activities of NF-κB and AP-I (two signaling pathways closely related to the expression of inflammatory factors) were also significantly inhibited (Figure 6B,C). The results suggest that DUSP5 might play an important role in innate immunity during SGIV infection.

### 3.7. E. coioides DUSP5 Promotes SGIV-Induced Apoptosis

It is well known that SGIV could induce typical apoptosis in FHM cells [21]. In this study, FHM cells transfected using pcDNA-3.1-3HA-DUSP5 for 24 h were infected with SGIV for 24 h. The effect of SGIV-induced apoptosis was detected by flow cytometry. The percentage of early apoptosis in control group cells was 31.1%, while that of pcDNA3.1-3HA-DUSP5 cells was 29.7%; and the percentage of late apoptosis in control group cells was 15.2%, while that of pcDNA3.1-3HA-DUSP5 cells was 20.7% (Figure 7A). As shown in Figure 7B, the pcDNA3.1-3HA-DUSP5 group significantly promoted late apoptosis compared to the control group (*p* < 0.05). The expressions of the apoptosis-related genes Bax and p53 were significantly upregulated, but the expression of Bcl-2 was significantly downregulated in the cells overexpressing DUSP5 during SGIV infection (Figure 7C). According to Western blotting, the protein level of caspase-3 in the cells of overexpressing DUSP5 (0.99) was higher than that of the control group (0.72), and the protein level of cleaved caspase-3 in the cells of overexpressing DUSP5 (1.08) was also higher than that of the control group (0.81), showing that overexpression of DUSP5 could increase the expressions of caspase-3 and cleaved caspase-3 (Figure 7D). The results indicate that DUSP5 could significantly promote SGIV-induced apoptosis in FHM cells.

## 4. Discussion

DUSP5 could be involved in many important biological processes, including immunity, cancer, cell proliferation, and cell autophagy [12,22,23]. In this study, the molecular characteristics of *E. coioides* DUSP5, its expression pattern, its effect on viral replication and virus-induced cellular regulation, and its effect on the innate immune response were explored.

The DUSP family shares a common phosphatase structural domain containing conserved Asp, Cys, and Arg residues forming the catalytic site [4]. As expected, *E. coioides* DUSP5 has the conserved Dusp super family domain. Analysis of the phylogenetic tree showed that DUSP5 was highly conserved between species, indicating that the function of DUSP5 in different species is similar; and *E. coioides* DUSP5 had the highest homology with DUSP5 of *E. lanceolatus*, suggesting that the roles of DUSP5 between *E. coioides* and *E. lanceolatus* might be more similar than other species. 

In mammals, DUSP5 belongs to the cytosolic DUSP family and is mainly distributed in the nucleus [6]. DUSP5 expression is significantly downregulated in patients with gastric or rectal cancer, and endoplasmic reticulum stress can affect the expression of DUSP5 via the PERK-CHOP pathway [24]. DUSP5 from marine fish *J. flounder* was most abundant in the skin and least so in the blood, and was activated by LPS (or Poly I: C) [3]. Similarly, *E. coioides* DUSP5 was mainly distributed in the nucleus. It was detected in all of the tissues, but was most abundant in the skin and least in the blood, suggesting that *E. coioides* DUSP5 might participate in various arrays of physiological reactions. *E. coioides* DUSP5 could be activated by SGIV infection, indicating that it might be involved in the development of SGIV infection. 

SGIV ICP18 and LITAF of are early viral genes [17,19]. The upregulated ICP18 can facilitate cell growth and viral infection, and activated LITAF can promote apoptosis and activate NF-kB signaling [17,19]. SGIV MCP and VP19 are late viral genes. MCP is one of the viral proteins, and VP19 plays a crucial role in cell migration [17,19]. In this study, overexpression of *E. coioides* DUSP5 reduced SGIV-induced CPE and significantly inhibited the expressions of ICP18, LITAF, MCP, and VP19, further indicating that *E. coioides* DUSP5 could inhibit SGIV replication. 

DUSP5 in mammals plays an important role in the regulation of inflammation [8,25]. Knockdown of DUSP5 can inhibit IL-1β-induced expression of inflammatory genes and activation of NF-κB and ERK signaling pathways, increasing the incidence of osteoarthritis [26]. DUSP5 could inactivate ERK1/2 through dephosphorylation, link TNF-αto pro-inflammatory genes in the nucleus, and reduce adipose tissue inflammation [27]. The expression level of DUSP5 was significantly reduced in SARS-CoV-2-infected patients [28]. In this study, SGIV infection could upregulate the expression of *E. coioides* DUSP5, and overexpression of *E. coioides* DUSP5 could inhibit the activity of NF-kB/AP-1 signaling pathways, reduce the expression of pro-inflammatory factors (IL-6, IL-8, and TNF-α), and reduce the SGIV-induced inflammatory response.

Apoptosis plays an important role in the evolution of organisms, the stability of their internal environment, and the development of multiple systems [29,30]. SGIV can induce typical apoptosis in FHM cells and enhance the formation of apoptosis by activating Caspase3 activity [21,31]. p53 is a key functional factor in initiating apoptosis in the cell cycle, and promotes apoptosis by activating Bax and inhibiting Bcl-2 [32,33]. In gastric cancer cells as well as colon cancer cells, overexpression of DUSP5 significantly promoted apoptosis [34,35]. In this study, overexpression of DUSP5 was able to significantly promote cell apoptosis; upregulate the expressions of caspase3, Bax, and p53; and downregulate the expression of Bcl-2 during SGIV infection, indicating that *E. coioides* DUSP5 regulates SGIV-induced cell apoptosis by regulating the expression of caspase3, p53, Bax, and Bcl-2.

In conclusion, *E. coioides* DUSP5 was characterized, and it was found to be distributed in all of the examined tissues. SGIV infection was able to activate *E. coioides* DUSP5. Overexpression of *E. coioides* DUSP5 significantly inhibited SGIV replication and reduced the expression of pro-inflammatory factors and the activities of NF-kB/AP-I, but promoted SGIV-induced apoptosis. These findings provide new insights into the molecules involved in SGIV infection, but also make an important contribution to understanding the pathogenesis of iridovirus. 

## Figures and Tables

**Figure 1 viruses-15-01807-f001:**
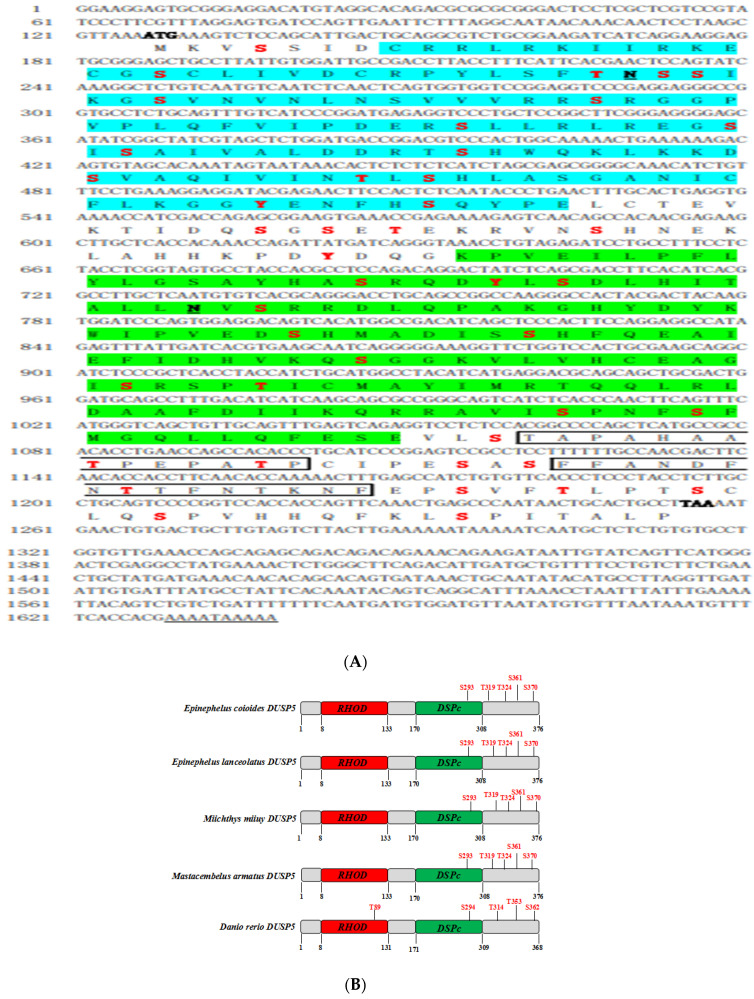
The full-length cDNA sequence, structural characterization, and phylogenetic position of *E. coioides* DUSP. (**A**) The start cordon (ATG) and the terminal codon (TAA) are shown in bold; the rhodanese homology domain (aa8-133) is shown in blue; the dual-specificity phosphatase catalytic domain (aa170-308) is shown in green; and the low-complexity region (aa312-325, aa333-347) is shown in a black box. The N glycosylation sites are bold black with an underline, and the 43 phosphorylation sites are in bold red font. Poly A is shown with an underline. (**B**) Comparison of different p38MAPK phosphorylation sites in different teleosts. The rhodanese homology domain is shown in red, the dual-specificity phosphatase catalytic domain is shown in green, and the red fonts indicate the phosphorylation sites of DUSP5 on p38MAPK. (**C**) Phylogenetic analysis of DUSP5 based on amino acid sequences. The phylogenetic tree constructed using the neighbor-joining (N-J) method with MEGA5.04 software. The branches were validated by bootstrap analysis with 1000 replications, which are represented by percentages in the branch nodes.

**Figure 2 viruses-15-01807-f002:**
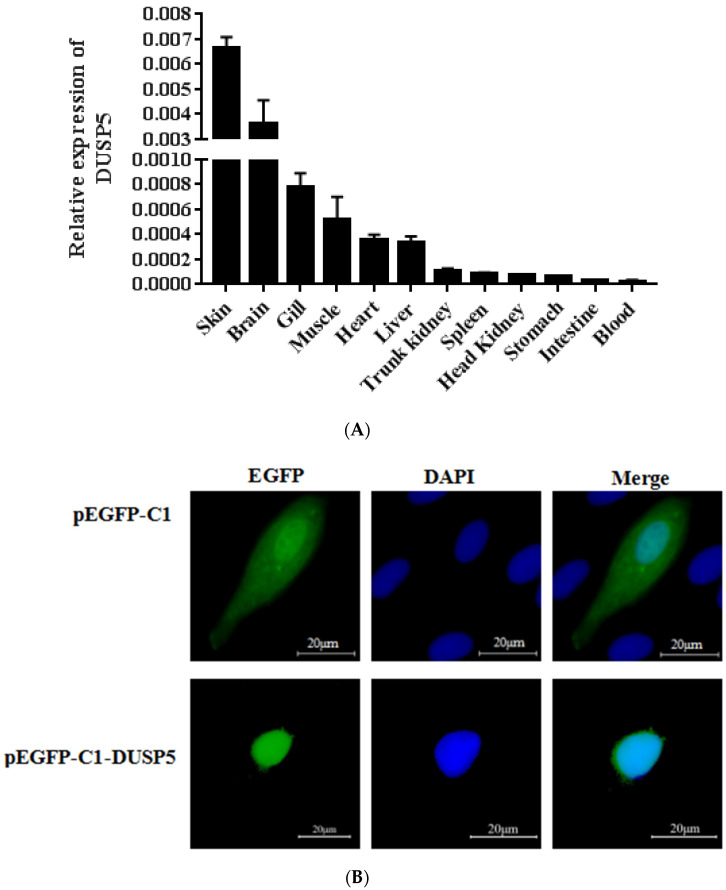
The distribution of *E. coioides* DUSP5. (**A**): The relative expression levels of DUSP5 in healthy *E. coioides*. β-actin was used as internal reference, and the data were calculated with the 2^−ΔΔCt^ method. All data are presented as mean ± SD, *n* = 3. (**B**): Subcellular distribution of *E. coioides* DUSP5. GS cells were transfected with an empty vector (pEGFP-C1) or a recombinant plasmid (pEGFP-DUSP5). The nucleus was stained by DAPI. The blue fluorescent markers represent the nucleus and the green fluorescent markers indicate the DUSP5 recombinant plasmid. Scale bars represent 20 μm.

**Figure 3 viruses-15-01807-f003:**
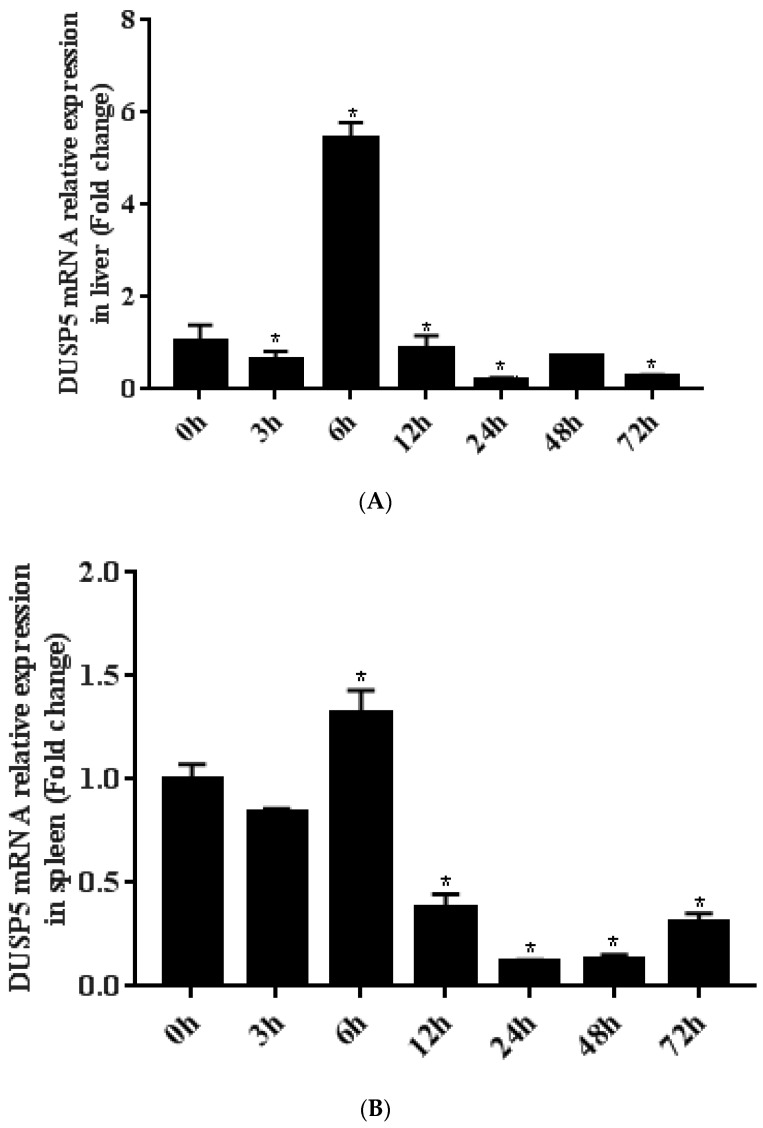
Expression of *E. coioides* DUSP5 response to SGIV infection in the liver (**A**) and spleen (**B**). The histogram showed the relative expression level of genes normalized by β−actin transcript and calibrated using the expression of DUSP5 at 0 h as 1. The DUSP5 mRNA expression levels at different times post−injection were determined using qPCR. Significant differences at each time point are indicated with * (*p* < 0.05). All data are presented as mean ± SD, *n* = 3.

**Figure 4 viruses-15-01807-f004:**
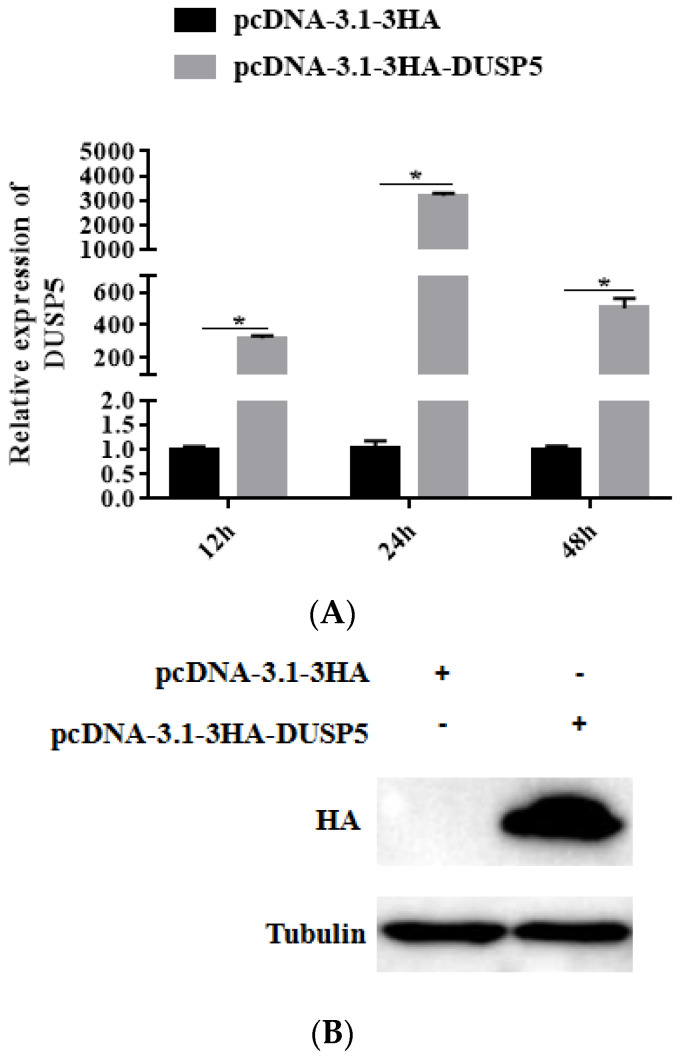
Efficiency of DUSP5 overexpression. Plasmid pcDNA3.1-DUSP5/pcDNA3.1-3HA was transfected into GS cells in 24-well plates, and the cells were collected at 12 h, 24 h, and 48 h for quantitative analysis and Western blotting. (**A**) The expression level of DUSP5 in DUSP5-overexpression cells at different times. (**B**) Expression of HA protein in cells transfected with pcDNA3.1-DUSP5/pcDNA3.1-3HA plasmid; β-tubulin was used as the internal control. Data are presented as mean ± SD, *n* = 3. Significant differences are indicated with * (*p* < 0.05).

**Figure 5 viruses-15-01807-f005:**
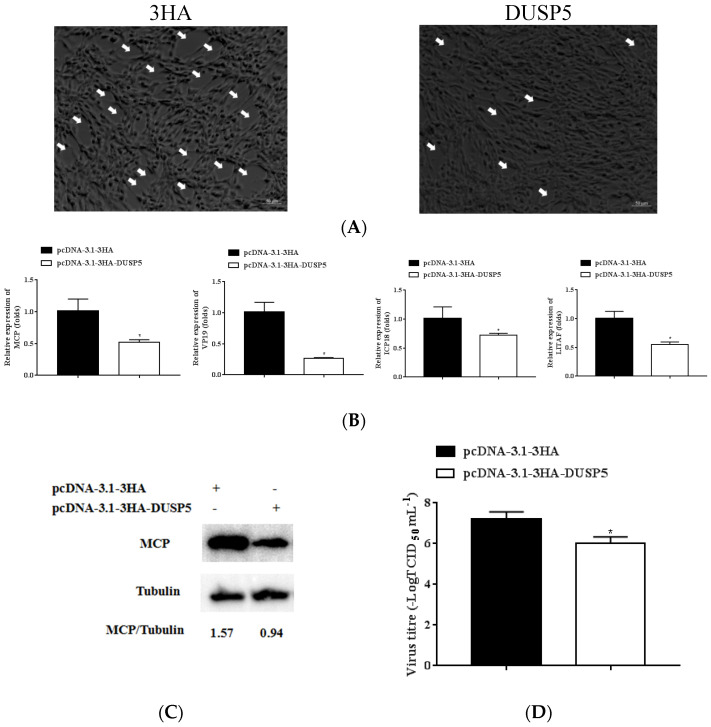
Overexpression of *E. coioides* DUSP5 inhibits SGIV infection. GS cells in 24-well plates transfected with the plasmid pcDNA3.1−DUSP5/pcDNA3.1-3HA for 24 h were infected by SGIV for 24 h, then collected. (**A**) The SGIV-induced CPE in GS cells. The white arrows indicate the formation of CPE caused by SGIV infection for 24 h. Scale bars represent 50 μm. (**B**) The expression level of four key SGIV genes (MCP, ICP18, VP19, and LITAF). (**C**) Western blot detection of the SGIV-MCP protein; β-tubulin was used as the internal control. (**D**) The viral titers in each group were measured using the TCID_50_ method. Data are presented as mean ± SD, *n* = 3. Significant differences are indicated with * (*p* < 0.05).

**Figure 6 viruses-15-01807-f006:**
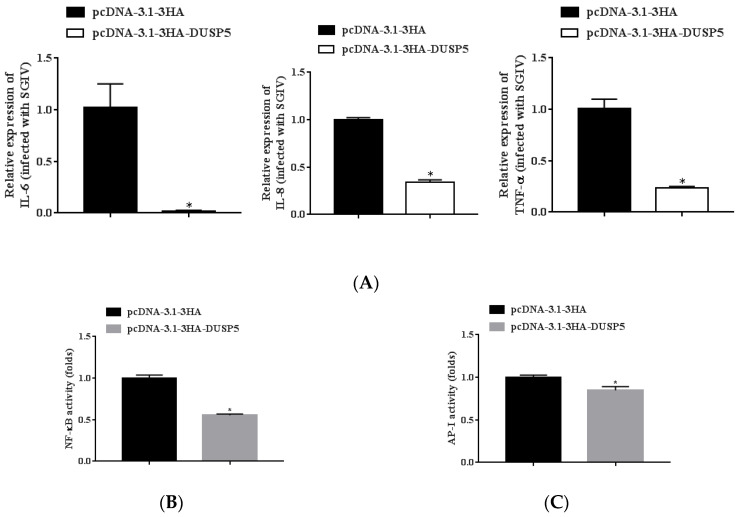
DUSP5 decreases the expression of immune-related factors. (**A**) qPCR was used to detect the expression levels of the pro-inflammatory cytokines (IL-6, IL-8, and TNF-α) in DUSP5-overexpressing GS cells infected with SGIV for 24 h. The relative luciferase activity of NF-κB (**B**) and AP-1 (**C**) was measured as a ratio of firefly luciferase activity to Renilla luciferase activity. All data were presented as mean ± SD, *n* = 3. Significant differences are indicated with * (*p* < 0.05).

**Figure 7 viruses-15-01807-f007:**
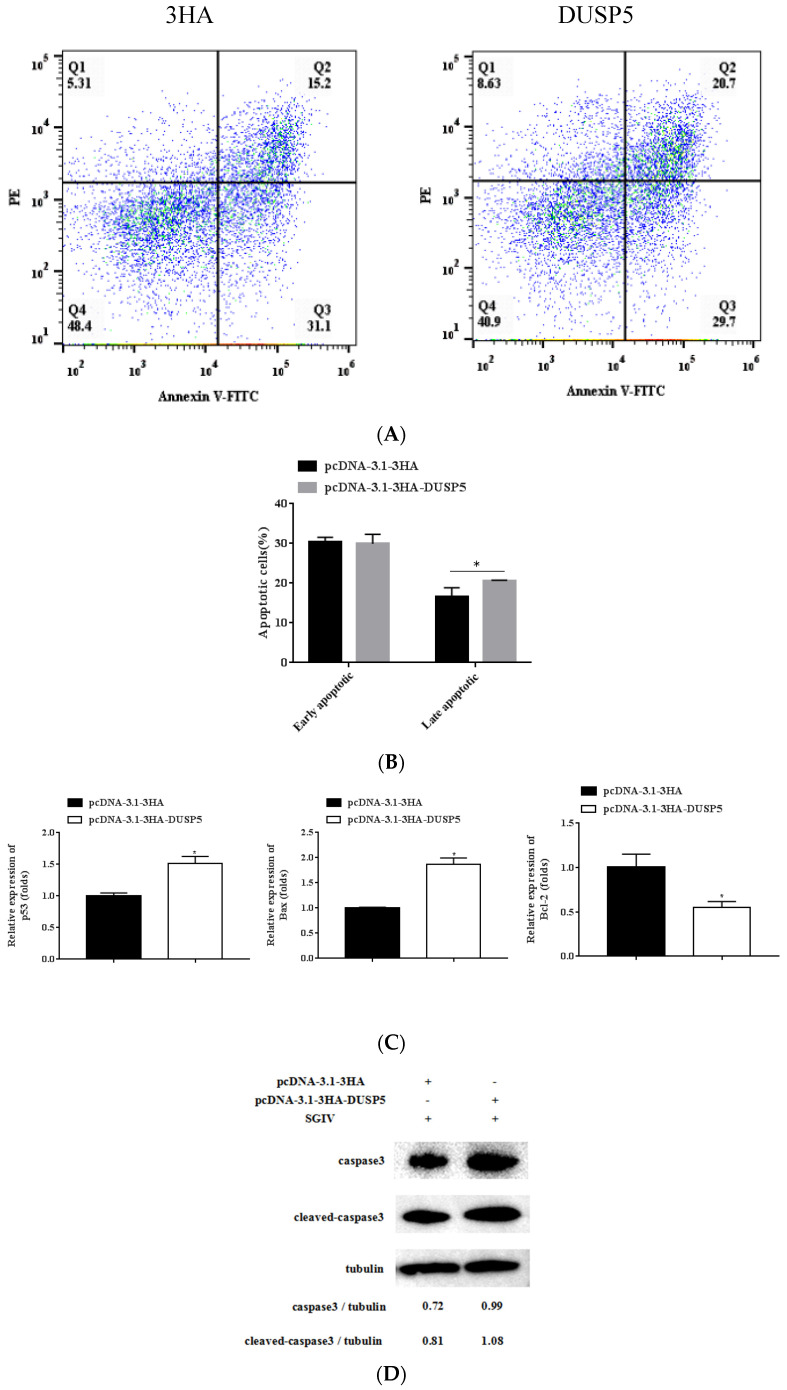
Overexpression of DUSP5 promotes SGIV−induced apoptosis. FHM cells transfected with pcDNA3.1−DUSP5/pcDNA3.1−3HA plasmid for 24 h were infected by SGIV for 24 h. (**A**) Annexin-V +/PI-cells (Q3) and annexin-V +/PI + (Q2) cells are early apoptotic cells and late apoptotic cells, respectively. Fragments of no cell structure (Q1) and normal FHM cells (Q4) were also exhibited. (**B**) This graph is based on the average of three parallel experimental results. (**C**) Three apoptosis-related genes (p53, Bax, and Bcl-2) were detected by qPCR. (**D**) Western blot detection of the proteins of caspase3 and cleaved caspase3; β-tubulin was used as the internal control. Data are presented as mean ± SD, *n* = 3. Significant differences are indicated with * (*p* < 0.05).

**Table 1 viruses-15-01807-t001:** Sequences of primers used in this study.

Primer Name	Sequence (5′ to 3′)
DUSP5-ORF-F	ATGAAAGTCTCCAGCATTGAC
DUSP5-ORF-R	TTAAGGCAGTGCAGTTATTGG
DUSP5-RT-F	TCCTGAAAGGAGGATACGA
DUSP5-RT-R	TCACTTCCGCTCTGGTC
DUSP5-pcDNA3.1-F	GCATCAGCGGAAAAGATGAAAGTCTCCAGCATTGACTGC
DUSP5-pcDNA3.1-R	ACTGTGCTGGATATCTTAAGGCAGTGCAGTTATTGGGCT
DUSP5-pEGFP-F	TACAAGTCCGGACTCATGAAAGTCTCCAGCATTGACTGC
DUSP5-pEGFP-R	GGTGGATCCCGGGCCTTAAGGCAGTGCAGTTATTGGGCT
MCP-RT-F	GCACGCTTCTCTCACCTTCA
MCP-RT-R	AACGGCAACGGGAGCACTA
LITAF-RT-F	GATGCTGCCGTGTGAACTG
LITAF-RT-R	GCACATCCTTGGTGGTGTTG
ICP18-RT-F	ATCGGATCTACGTGGTTGG
ICP18-RT-R	CCGTCGTCGGTGTCTATTC
VP19-RT-F	TCCAAGGGAGAAACTGTAAG
VP19-RT-R	GGGGTAAGCGTGAAGACT
IL-6-RT-F	CTCTACACTCAACGCGTACATGC
IL-6-RT-R	TCATCTTCAAACTGCTTTTCGTG
IL-8-RT-F	GCCGTCAGTGAAGGGAGTCTAG
IL-8-RT-R	ATCGCAGTGGGAGTTTGCA
TNFα-RT-F	GTGTCCTGCTGTTTGCTTGGTA
TNFα-RT-R	CAGTGTCCGACTTGATTAGTGCTT
β-actin-RT-F	TACGAGCTGCCTGACGGACA
β-actin-RT-R	GGCTGTGATCTCCTTCTGCA
Bax-RT-F	TGTGCGACCCAAATACCAAGAGG
Bax-RT-R	AAGTAGAACAGTGCAACCACCCTGC
p53-RT-F	GGAGGAAAACAGCACCAAGACGC
p53-RT-R	CCACGAACATGCAGAACAAACACG
Bcl-2-RT-F	ATGAACAAAGAAGTAGATTGGGTCG
Bcl-2-RT-R	GTGAGATGAGTAAGGAAGGGATGA

## Data Availability

All data generated or analyzed in the current study are included in this published article. Please contact the corresponding author for details.

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
