# Peer review of "The Role of Epinephelus coioides DUSP5 in Regulating Singapore Grouper Iridovirus Infection"

_viruses, 2023, doi:10.3390/v15091807_

Round 1
Reviewer 1 Report
The manuscript viruses-2544265 describes the role of the DUSP5 gene in Epinephelus coioides in Iridovirus infection. The paper represents a key to understanding the genesis and epidemiology of grouper iridovirus infection, generating results that can be used in other similar viral infections.
First of all, it should be noted that the authors did not follow the formatting instructions of the bibliography and citations in the text: an accurate and careful revision of the entire manuscript is therefore required.
The introduction is sparse and not very incisive; news on the three subjects of the work is missing or almost non-existent, in order to better detail the paper and to provide users with a basis on which to contextualize the topic.
In the second line of page 1, the citation Lin et al., 2020 is missing in the references or the date inserted is incorrect (the date 2006 of the paper has been inserted in the references).
The materials and methods, regarding subchapter 2.1, are unclear: it is important to better explain the experimental design; moreover, all the descriptions of the culture conditions used (T°, O2, etc.) are missing. finally, the data relating to the ethical statement and the legislation on animal experimentation of reference if necessary are missing.
In the results, on page 5, sixteenth line of subchapter 3.1, eliminate the full name of the species Epinephelus, replacing it with E.; in subchapter 3.2, sixth line, delete the second intestine (repetition). Figure 2 uniformly format the characters used.
The discussion could be better articulated. Review the references as requested above.
For these reasons, in my opinion, this manuscript must undergo major revisions in order to proceed with the publication process.
Author Response
1. First of all, it should be noted that the authors did not follow the formatting instructions of the bibliography and citations in the text: an accurate and careful revision of the entire manuscript is therefore required.
We thank the referee for his/her suggestion, and the references and the citations have been revised according to the magazine. Please see full length of the Ms. Thank you very much.
2. The introduction is sparse and not very incisive; news on the three subjects of the work is missing or almost non-existent, in order to better detail the paper and to provide users with a basis on which to contextualize the topic.
We thank the referee for his/her suggestion, and the introduction has been revised. Please see the section of introduction. Thank you very much.
3. In the second line of page 1, the citation Lin et al., 2020 is missing in the references or the date inserted is incorrect (the date 2006 of the paper has been inserted in the references).
We thank the referee for his/her suggestion, and the references have been revised. Thank you very much.
4. The materials and methods, regarding subchapter 2.1, are unclear: it is important to better explain the experimental design; moreover, all the descriptions of the culture conditions used (T°, O2, etc.) are missing. finally, the data relating to the ethical statement and the legislation on animal experimentation of reference if necessary are missing.
We thank the referee for his/her suggestion, and the section 2.1 has been revised. Thank you very much.
5. In the results, on page 5, sixteenth line of subchapter 3.1, eliminate the full name of the species Epinephelus, replacing it with E.; in subchapter 3.2, sixth line, delete the second intestine (repetition). Figure 2 uniformly format the characters used.
We thank the referee for his/her suggestion, and Epinephelus (section 3.1) has been repliced with E.; the second intestine has been deleted; the format in Figure 2 has been uniformed. Thank you very much.
6. The discussion could be better articulated. Review the references as requested above.
We thank the referee for his/her suggestion, and the discussion has been revised. Thank you very much.
Reviewer 2 Report
I do not have access to line numbering, thereby making specific points more difficult
What is an "economically marine fish" - is a word missing? (last paragraph, introduction). Apart from this "beauty error", the introduction appears adequate.This is a fine manuscript, that should and I find few things to criticise. The materials and Methods are solid, and the results in my view, merit publication. My only general comment is that the manuscript is long and relatively complex, thus it would probably benefit from adding a short Conclusions section.
Author Response
1. I do not have access to line numbering, thereby making specific points more difficult
We thank the referee for his/her suggestion, and the line number has been added. Thank you very much.
2. What is an "economically marine fish" - is a word missing? (last paragraph, introduction). Apart from this "beauty error", the introduction appears adequate.This is a fine manuscript, that should and I find few things to criticise. The materials and Methods are solid, and the results in my view, merit publication. My only general comment is that the manuscript is long and relatively complex, thus it would probably benefit from adding a short Conclusions section.
We thank the referee for his/her suggestion, and "economically marine fish" has been described (line 64-65); and the discussion has been revised. Thank you very much.
Round 2
Reviewer 1 Report
The authors have reviewed the viruses-2544265 manuscript as requested by the referees. I have not highlighted any problems in the new draft; therefore this paper can be started for publication.